# Barriers and Facilitators to Mental Health Help-Seeking and Experiences with Service Use among LGBT+ University Students in Chile

**DOI:** 10.3390/ijerph192416520

**Published:** 2022-12-09

**Authors:** Marcelo A. Crockett, Vania Martínez, Patricio Caviedes

**Affiliations:** 1Escuela de Salud Pública, Facultad de Medicina, Universidad de Chile, Santiago 8380453, Chile; 2Millennium Nucleus to Improve the Mental Health of Adolescents and Youths (Imhay), Santiago 8380455, Chile; 3Millennium Institute for Research in Depression and Personality (MIDAP), Santiago 7820436, Chile; 4Centro de Medicina Reproductiva y Desarrollo Integral del Adolescente (CEMERA), Facultad de Medicina, Universidad de Chile, Santiago 8380455, Chile

**Keywords:** LGBT+, mental health, help-seeking, service use, university students

## Abstract

Barriers limiting access to mental health care for lesbian, gay, bisexual, trans, and other sexual and gender minority (LGBT+) university students have not yet been explored in depth. The aim of this study was to explore the barriers and facilitators to mental health help seeking and experiences with service use among LGBT+ university students. Participants were 24 LGBT+ students between 18 and 23 years of age from a university in Chile. Individual semi-structured interviews were conducted and analysed using thematic content analysis. Multiple barriers and facilitators influence mental health help-seeking of LGBT+ students, with some of these barriers being explicitly related to LGBT+ issues (e.g., fear of discrimination or accessing specialised services). Perceived effectiveness of services was closely related to access safe/affirming care. Trans students reported more barriers to help-seeking and negative experiences with professionals than their cisgender peers. Perceptions of university mental health services as safe spaces for LGBT+ students were related to a positive perception of the university regarding LGBT+ issues. Knowing the factors that either hinder or facilitate help-seeking and characterising service use experiences in this population is useful for improving access to mental health services and for the development of policies that promote affirmative care for LGBT+ people.

## 1. Introduction

Mental health problems are highly prevalent in university students [1,2]. Nearly one-third of students have had a mental health problem during the past 12 months, with major depression and generalized anxiety disorder being the most common issues [1]. Among university students, lesbian, gay, bisexual, trans, and other sexual and gender minority (LGBT+) students exhibit worse mental health outcomes than their heterosexual and cisgender (HC) peers around the world [1,3,4,5,6] as well as in Chile [7].

In addition to the normative challenges of emerging adulthood [8], LGBT+ youths face other challenges, such as the disclosure of their sexual orientation and gender identity (SOGI), gender transition, unsafe university contexts, and limited access to inclusive university health services [9,10,11]. According to the minority stress model, factors such as stigma, prejudice and discrimination against sexual and gender minorities create hostile and stress-generating social environments, that may explain the higher prevalence of mental health problems in LGBT+ populations [12,13].

Several studies indicate that LGBT+ university students receive more mental health counselling or treatment than their HC peers [5,14,15,16], both on and off university campuses [17]. Nevertheless, they report more barriers to accessing mental health services than their peers [14]. To understand why young people who need mental health treatment do not receive it, researchers have focused on studying the barriers or factors that might negatively affect help-seeking [18] and, to a lesser extent, the facilitators or factors that might motivate help-seeking [19].

Little is known about barriers and facilitators to mental health help-seeking in LGBT+ university students. Studies with general LGBT+ population samples have found a range of specific barriers to mental health help-seeking, such as high levels of internalized sexual stigma [20], fear of being discriminated against, experiences of discrimination or a lack of sensitivity to LGBT+ issues among providers [21].

Once they have accessed university mental health services, it has been reported that trans youth are usually not called by their updated name, misgendered, or asked irrelevant questions about their gender identity by some professionals [22]; furthermore, professionals with knowledge about trans issues are scarce [10]. In the general LGBT+ population, it has been observed that heterosexist and discriminatory behaviours by providers, low availability of affirmative services [23], microaggressions based on sexual orientation (e.g., assuming that sexual orientation is the cause of all problems, avoiding or minimizing the sexual orientation of clients) [24] and a lack of training in trans issues [25] can constitute barriers to the provision of effective and culturally sensitive mental health treatment for LGBT+ people, which can lead to early treatment dropout [26]. In contrast, positive attitudes towards sexual orientation and gender identity by professionals [27] and the provision of affirming LGBT+ services (culturally appropriate and sensitive approaches to working with LGBT+ people [28]) are associated with greater perceived satisfaction with mental health services [25,26].

Universities generally have various health services for their students, which offer psychological treatment and other services depending on demand and resource availability [29]. Given their universal and low-cost nature, understanding the processes related to access to university services could be useful in identifying other factors that influence access beyond equity in access and cost [30]. For instance, one study reported that LGBQ students preferred to use off-campus services possibly to obtain affirmative care, which may not be available or advertised in university service offerings [14]. In Chile, although there is no published data about how many students access university mental health services, it appears that the number has been increasing over the years [31].

Identifying the factors that either hinder or facilitate mental health help-seeking could help explain the disparities in this population’ use of services, which could be useful for policies and interventions aimed at promoting help-seeking. This is especially relevant in the Latin American context where there is less research on the LGBT+ population [32,33] and less progress has been made to protect rights through public policies [34]. In Chile, despite some legislative advances in the last years, discrimination and violence against LGBT+ people are highly prevalent [35] and negative attitudes towards LGBT+ people are still an issue. For example, a recent study found that 34% of public health workers had homophobic attitudes [36]. This is especially relevant since 12% of young people between 15–29 years report an LGBT+ identity in Chile [37]. Furthermore, stigma towards mental illness remains as an important challenge, particularly towards people with severe mental disorders [38]. Taking the above into account, the aim of this study is to explore barriers and facilitators to mental health help-seeking and examine LGBT+ university students’ service utilization experiences in Chile.

## 2. Materials and Methods

### 2.1. Participants

In this qualitative study, 24 participants were recruited from The WHO World Mental Health International College Student initiative (WMH-ICS) at one of the participating universities in Chile. A purposeful sample was collected using the following inclusion criteria: being a first or second year student, being 18 years of age or older, and identifying as lesbian, gay, bisexual, trans, or another sexual orientation and/or gender identity different from heterosexual and cisgender. The recruitment process considered the diversity of the participants’ experiences based on the different types of SOGI reported in the WMH-ICS survey.

The characteristics of the participants are shown in Table 1. All participants had used mental health services during their lifetime, except for one participant (E1) who used a psycho-pedagogical university service. Most of the students (n = 23) had used private outpatient mental health services, 8 had used public outpatient mental health services, 5 had used university mental health services, and 4 had used LGBT+ specialised services. At the time of the interview, 20 students were receiving mental health care. The most common reasons for consulting were mental health problems. Fifty-four percent of the participants had public health insurance and the rest had private health insurance. Those who had public health insurance also used private services. One third of the students came from public schools, one third from state-subsidized schools, and one third from private schools. Two thirds of the students were from the Santiago area (Chile’s capital) and one third came from other regions of the country. The participants came from 19 university courses and 10 different faculties or institutes of the university. This information was not included in the table to protect the anonymity of the participants.

Students were recruited from one public university to deepen the analysis in a particular context. This university is characterized by: (1) a diverse context due to the existence of multiple faculties, institutes, and campuses, (2) student diversity, (3) the presence of several university mental health services, one centralized for all students and other services in some of its faculties, (4) the presence of two units focused on the inclusion of LGBT+ students and the prevention of discrimination against them at the university, and (5) the existence of policies to promote the use of updated names and non-discriminatory treatment based on gender identity.

### 2.2. Data Collection Technique

Semi-structured individual interviews were conducted with the participants. This technique was chosen because it is flexible enough to adapt to the characteristics of the conversation and to address contents not included in an interview script [39]. Based on the literature on the use of health services and barriers to help-seeking in LGBT+ university student populations [22,40,41], as well as in general LGBT+ populations [23,42], five topics were defined to be addressed in the interview: (1) Experiences with the use of mental health services in and out of university (including barriers and facilitators that influence help-seeking); (2) Willingness to use mental health services in the face of current or future mental health problems; (3) Desirable characteristics of professionals during help-seeking; (4) Recommendations for improving mental health services; (5) Experiences as an LGBT+ university student. The latter topic was incorporated with the purpose of exploring the influence of contextual factors on the use of university health services. Semi-structured interviews were conducted until data saturation was achieved (i.e., the last interviews repeated what had been found in previous ones) [43]. Each participant was interviewed once and the interviews lasted an average of 60.9 min.

### 2.3. Procedure

First, authorization was obtained from the university and the Faculty Human Research Ethics Committee for this study. Then, email invitations were sent to students who had previously participated in the WMH-ICS study in Chile by the second author (the principal investigator of the WMH-ICS project in Chile), explaining the objective of the study and that the recruitment was focused on LGBT+ students. Individual interviews were conducted by the first author with each participant over video calls due to public health restrictions adopted in response to the pandemic and to avoid SARS-CoV-2 transmission. Students signed informed consents via an online form prior to participation. Interviews were audio-recorded, transcribed verbatim, and reviewed for accuracy. Personal data and names were changed to protect the anonymity of the participants. Whenever an unmet mental health need was detected, an assisted referral was made at the end of the interview. This was done twice. Participants were contacted and interviewed between July 2021 and October 2022.

The first and third authors are psychologists with master’s degrees in research in psychology and inequalities and social science, respectively. The first author is also a PhD student in public health and this study is part of his doctoral thesis, which focuses on exploring mental health inequalities among LGBT+ university students. The second author is a psychiatrist with a PhD in psychotherapy and the first author’s thesis advisor. All three authors have previously worked with LGBT+ youth in clinical settings. Their research interests are focused on the mental health of young people and the promotion of safe spaces for LGBT+ youth in mental health settings. At the beginning of each interview, the first author introduced himself, the objective of the study, and the reason for conducting the study.

### 2.4. Analysis

The interviews were analysed using inductive thematic content analysis, i.e., the categories of analysis were derived from the data and not from a previously defined theory [44], given the exploratory and descriptive scope of the study. Open coding was used to identify all possible themes emerging from the interviews. All interviews were coded independently by the first and third authors of this study. Differences were compared and discussed until a consensus was reached. The codes were grouped into categories according to their content, and each category was described and named to reflect its main findings. MAXQDA 2020 software package was used for the analysis.

To ensure the fulfilment of quality criteria in qualitative research (credibility, transferability, dependability and confirmability), the following measures were adopted: (1) prolonged meetings were held to build trust with the participants due to the sensitive and personal nature of the information requested; (2) data collection, coding, and analysis were carried out concurrently, which served to constantly review the coding done; (3) researcher triangulation was incorporated into the analysis process described above; (4) at the end of the interview, participants were given the opportunity to correct or clarify the themes addressed and add new information; (5) a detailed description was generated to specify the design, participants, and context of the study; (6) the results were presented at meetings to be discussed by other researchers [45].

## 3. Results

The results were 150 codes, organized into eight categories, which are described below according to the temporality of mental health help-seeking. Four were related to mental health help-seeking ((1) barriers to and (2) facilitators of help-seeking, (3) desirable characteristics of mental health professionals, and (4) willingness to use university services) and three were related to experiences with service use once they had accessed care ((5) barriers to and (6) facilitators of effective mental health care, and (7) disclosure of sexual orientation and gender identity to mental health professionals). The final category (8) captures the participants’ recommendations for professionals and services during help-seeking and service use.

### 3.1. Barriers to Mental Health Help-Seeking

Most participants reported various barriers that prevented or delayed help-seeking. These barriers were classified according to who or what affected help-seeking: individual (related to personal attitudes, beliefs, or experiences), social (related to family or peer attitudes and beliefs), and service (related to more structural aspects of mental health services). Each student reported one or more barriers, except for one participant who reported no difficulties in accessing mental health care (E9).

Regarding individual barriers, participants reported some problems explicitly related to LGBT+ issues such as fear of being discriminated against by health professionals, confidentiality concerns regarding their parents finding out their SOGI, and negative past experiences with mental health professionals (e.g., discrimination, interruption of care after disclosure of gender identity). These barriers were reported by seven trans students (male, female, non-binary, and gender fluid) and one female cisgender student. Overall, participants reported a wide range of barriers related to attitudes about coping with mental health problems. For example, minimizing the problem, wanting to deal with the problem on their own, fear of finding out that it was something worse, not wanting to bother other people with their problems, and that males should hide their feelings and not seek help. Also, students reported barriers related to beliefs or knowledge about mental health care, including negative assumptions about mental health care, trust in the available mental health care services, negative experiences with services (personal or other people’s), a lack of knowledge about how to access online therapy, not knowing when it is the right time to seek help, or fear that the professional will minimize their problems. They also reported barriers related to a lack of time due to heavy academic workloads and psychological symptoms preventing them from seeking help (e.g., insufficient energy and motivation).

In relation to social barriers, some students identified their family, especially when they do not have the age/autonomy to access mental health services on their own. They reported experiences in which their family minimized distress and determined that they did not need mental health care. They also identified negative beliefs about mental health care in their family and peers at school.

In relation to services, two non-binary, one gender fluid, and one lesbian cisgender student (all of whom had public health insurance), reported the low availability and high cost of professionals who specialise in LGBT+ issues as well as the denial of mental health care based on trans gender identity. Other barriers reported were the low availability of non-specialised mental health services (e.g., lack of available appointments, long waits to access public health care, high demand in the private health care due to the pandemic) and the high financial cost of private mental health care. In the following quote, a student reports a combination of barriers: low availability and high cost of mental health care by professionals specialised in LGBT+ issues:

*I knew that, for example, a psychologist with a gender approach would have been very, very cool, but they don’t have appointments like never and they are a bit more expensive than usual. So, I really, I wanted to, but I knew that I couldn’t have it because it’s complicated* (E8).

### 3.2. Facilitators of Mental Health Help-Seeking

Participants mentioned a range of factors or people who contributed positively to help-seeking. Like barriers, facilitators were also classified as individual, social and service-related. Each student described two or more different facilitators, except for one student who had been using mental health services since childhood (E3).

The individual facilitators reported are the recognition of mental health distress, positive attitudes towards mental health care, seeking mental health information online, and having interrupted mental health treatments in the past. At the social level, students highlighted parental support and aid in the help-seeking process. After the students recognized their own need for care, they went to their parents (especially to their mothers) to talk about mental health help-seeking. In some cases, the family initiates the search for help, for example, when the student refuses to seek professional care and, as reported by two trans participants, because of concerns about their gender expression. Partners, siblings, and friends can also play an active role by recommending seeking professional help when they identify the participants’ distress. In terms of services, the facilitators identified were the recognition of mental health problems by education and health professionals, receiving care from professionals who are known or recommended by people close to them, availability of information about the professionals or how to access care, the existence of formal agreements between the university and external psychological care centres, reduced fees or being able to afford private services, and the availability of mental health professional appointments. In the following quote, a student reports how she recognized her discomfort and how her mother supported her to start seeking help.

*I was feeling very bad very often, I wanted to cry a lot, so I told my mum “Mum, I’m feeling very bad. It doesn’t seem normal to me, I think that, maybe, I need to go to a psychologist” and she told me “Okay, dear” and she got me one [an appointment] there* (E14).

### 3.3. Desired Characteristics of Mental Health Professionals during Help-Seeking

When seeking mental health care, participants reported several characteristics of professionals and services that they would look for. Most of the students highlighted the positive attitude of professionals towards LGBT+ people. Specifically, participants reported preferring young and female professionals, whom they are considered to be more open-minded. Two cisgender students (one male and one female) mentioned that they did not care for the professional’s gender, while one cisgender male student mentioned that he would prefer a male professional. In addition, students reported that they would look for professionals with knowledge of LGBT+ issues, especially trans students (male, female, non-binary, and gender fluid). However, there were differences in the degree of knowledge expected from professionals. Some participants only expected them to have general knowledge that would promote respect and non-discrimination. In contrast, one group of students reported preferring professionals specialised in LGBT+ issues, while two students would prefer a professional who is part of the LGBT+ community because they would be more likely to have a better understanding of the participants’ SOGI. In this regard, some students made a distinction, noting that they intended to seek specialised professionals to address not only issues regarding their SOGI, but also other mental health issues:

*It had to be a queer space, but it didn’t have to be the only goal because I don’t go to the psychologist because I am queer, I go to the psychologist because I have mental health problems* (E16).

Other desired characteristics included affordable cost, in-person care, comfort in the relationship with the professional, geographical proximity to home, availability of online information about the professional, whether the professional provides care through their health insurance, the professional’s therapeutic approach, whether they had previously met the professional, the professional’s political position, and the type of mental health professional (e.g., psychologist or psychiatrist).

### 3.4. Barriers to Effective Mental Health Care

Once they had started receiving mental health care, participants commented on various barriers to its effectiveness, which meant that this care was not positively contributing to their well-being. Some of these barriers are explicitly related to LGBT+ issues. All these barriers can have negative effects during mental health service use, affecting trust and relationships with the professionals, impacting on the quality of the services delivered, or increasing the mental distress of the participants.

The barriers during the use of services involve discrimination based on SOGI and poor knowledge of LGBT+ issues. Participants reported experiences such as: denial of their SOGI (e.g., bisexuality is not real) or completely avoiding it after disclosure, not respecting client’s pronouns or names, making prejudiced comments towards LGBT+ people, being outed by the professional, discontinuation of the therapy by the professional after disclosure, questioning the client’s desire to receive cross-sex hormone therapy, or reducing all problems to the fact of being trans. Most of these negative experiences during service use are reported by trans, non-binary, and gender fluid students. None of these experiences were reported by male cisgender students. The following quote illustrates an experience related to the denial of a student’s gender identity.

*In fact, one day I remember I was talking to her [the psychologist] and she told me something like “if I were to see you on the street, I would say that you are an alternative woman”. Like, completely annulling my gender, so... it has been very uncomfortable. In fact, the last time I saw her in person (I had already been with her for more than a year) she realized that I was not a trans man, but rather, that I was a non-binary person, even though that I had told her that all year* (E16).

Five trans and gender fluid students mentioned that they received psychological care due to their gender identity, while other two students received psychological care due to their sexual orientation. Mental health help-seeking was motivated by parents due to gender expression in childhood and to access cross-sex hormone therapy in adolescence. One student reported that he received psychological counselling at school to be able to wear clothes in accordance with his gender expression. Another student received psychological care to prevent same-sex expressions of affection at school. Especially the experiences of care during childhood and in school could be conceived as sexual and gender identity change efforts, in which professionals approached the participants’ SOGI in a pathological way and made efforts to suppress it (for example, by prohibiting same-sex expressions of affection or recommending a trans student to socialize with boys to adopt “male codes”).

Other barriers are related to professionals and health services. They were also mostly reported by trans students and cisgender female students. Regarding professionals, young people reported several ways in which professionals’ comments caused discomfort to them and could have affected trust. For example, professionals minimized their distress or did not pay attention to their symptoms, repeatedly asked questions about non-relevant topics, did not explain what the medication was for, did not give a diagnosis, used a religious approach during psychological therapy, and caused confidentiality issues by having parents participate in the young people’s therapy (for example, when professionals treated adolescents and then talked to their parents). In terms of services, they reported problems with the frequency of care and with the decision to prescribe pills only if they stayed in psychological therapy, which was evaluated as unhelpful.

### 3.5. Facilitators of Effective Mental Health Care

Students reported a variety of factors that positively contributed to the quality and effectiveness of mental health services. Some of these facilitators of effective mental health care are explicitly related to LGBT+ issues. These facilitators relate to establishing a safe space for LGBT+ people, i.e., a space of respect, free from discrimination and prejudice. Some actions that contribute to this are: respecting names and pronouns (especially for trans students), a positive attitude of the professional towards LGBT+ people, not assuming the SOGI of people, rectifying mistakes with pronouns or making an effort to use neutral pronouns, not discriminating against or judging others based on SOGI, and having knowledge about LGBT+ issues. The latter is intended to avoid having to explain what gender identity is to professionals and to keep professionals from reducing all problems to the client’s SOGI. In the following quote, a student reflects on how respect and therapy effectiveness are closely related.

*[In therapy] my pronouns are respected and if at some point she kind of gets confused, she corrects herself. It is also a very important point for me. And well, I mean, no, I haven’t felt, I haven’t felt discriminated against because of that. [And how do you feel about the therapy process that you’re going through?] I think they go very much hand in hand, I mean, if I felt discriminated against, I wouldn’t feel like I was making much progress in therapy either, I would feel in a negative way* (E4).

Other facilitators concern the relationship with professionals and aspects related to care. For example, participants highlighted having a close and positive relationship with mental health professionals, the possibility of establishing a more horizontal relationship with professionals, and generational proximity, which contributes to a fluid communication. Other aspects, which may be particular to each student, are less frequently mentioned. These include: the professional’s validation of the distress that they are experiencing, reaching a mutual agreement with respect to the frequency of care, efforts from the professional to adapt to the participant’s availability or needs, being encouraged to engage in activities outside of therapy, and receiving explanations from the professional about what medications are for.

### 3.6. Disclosure of Sexual Orientation and Gender Identity to Mental Health Professionals

Disclosing or talking about SOGI in mental health care was a relevant aspect for the participants. Some expressed the desire to talk about their SOGI as they had some concerns and felt that it would be useful to discuss it with a professional. In this regard, one participant reflected on the importance of her sexual orientation in therapy and how it affects several aspects of her life.

*In the sense that, of course, that it is my [sexual] orientation, but it is also in my interest. I don’t know, when they talked to me about... about my ideals for therapy, I was asked a lot about that, or what are your hobbies or what you are passionate about. So that’s where [sexual] orientation meets my ideals, my hobbies and all that stuff, which is like, I don’t know, I like to read a lot about, I don’t know, queer theory or things like that, or my ideals are, I don’t know, I fight for the rights of the LGBT community. And of course, it wouldn’t be like that if I wasn’t part of the LGBT community* (E24).

More than half of the participants reported that they talked about these issues comfortably with mental health professionals, which was facilitated by the positive attitude of the professionals towards LGBT+ people, the trust that they place in the professional, or receiving autonomous mental health care, with no parental involvement. Individual aspects of the participants were also reported to have an influence and included having disclosed their SOGI to their family and friends or feeling comfortable about their SOGI. In the following quote, a participant comments on the reasons why she has been able to talk about her sexual orientation in the current therapy, as opposed to her previous therapy.

*One thing that’s been different, for a start, is the fact that I’m older and much more comfortable in my sexuality. That gives me a lot of confidence and I have a support network. Apart from that, there’s the fact that, because I’ve spent a lot of time with this psychologist, I’ve been able to build up a lot more confidence. Plus, the fact that I’m the only one seeing this psychologist and my family is not involved in my treatment. That has given me a sense of security that allows me to talk about it* (E6).

Three participants reported that they spoke uncomfortably about their SOGI in therapy because of negative situations in relation to disclosure. For example, a participant’s mother disclosed the participant’s sexual orientation to the professional, a professional questioned the student’s bisexuality, some professionals omitted the topic after disclosure, and a professional offered to treat the participant’s gender identity with therapy.

For their part, some participants had not talked about their SOGI in therapy due to personal factors such as preferring to discuss other issues, fear of discrimination, not having disclosed their SOGI to their family, negative attitudes, or insecurity about their gender identity. Other factors related to the professionals or health care were reported, such as the practitioner not asking questions about the participant’s SOGI, sexist comments, or concerns about confidentiality especially in online therapy, as they feared that others in the household might overhear them talking about their SOGI.

### 3.7. Willingness to Use University Mental Health Services

More than half of the participants showed willingness to use university mental health services, although only five had done so (four of them with public health insurance). To a lesser extent, students were also uncertain or held negative views, e.g., due to negative comments about these services. Some students reported not using university mental health services because they were using external services, or preferred private services, while others reported that the services may be more useful for students with more needs than them. Some students perceived university services as trustworthy because they associated them with the prestige of the institution. Also, some students perceived that these services could have a positive attitude towards LGBT+ people, possibly due to the general perception that the university is a safe environment for this population. In the following quote a student gives an account of her perceptions of how university mental health services might treat LGBT+ students.

*I feel that most of the community here in this faculty and campus is very attuned to the issue, so I think that people from the [LGBT+] community would be well received* (E24).

One of the barriers to using these services was that students were largely unaware of them despite having received information about them in induction talks or e-mails. The participants also identified the limited number of professionals as a problem, with one student reporting access difficulties because contact with the service must be established by email.

### 3.8. Recommendations for Professionals and Services Regarding LGBT+ Students

The recommendations made by the students covered three main elements: respect and non-discrimination, knowledge of LGBT+ issues, and general aspects of mental health service provision. In terms of respect and non-discrimination, the participants made several recommendations, such as asking the client’s name and pronouns (which was especially important for trans and cisgender female students), avoiding SOGI-based judgment, identifying the service as a safe space, and preventing discriminatory behaviour from professionals. Examples of these recommendations include not denying the participant’s SOGI and prohibiting sexual orientation and gender identity change efforts (also called “conversion therapy”) or openly LGBT-phobic professionals.

Knowledge of LGBT+ issues, one of the most frequent recommendations, involves knowing basic elements that could improve the quality of services for LGBT+ people. In this regard, nine trans (male, female, non-binary, and gender fluid) and two male cisgender students recommended that professionals must have training in LGBT+ issues. Specific recommendations about LGBT+ knowledge were: not assuming people’s SOGI, not expecting binary gender expression, using neutral language when referring to other people (e.g., saying partner instead of boyfriend or girlfriend), proposing to talk about SOGI in the participant’s own time, not generalizing the LGBT+ experience, and not reducing all problems to LGBT+ identity.

With respect to more general aspects of mental health service provision, the participants recommended facilitating access for more disadvantaged LGBT+ people and ensuring privacy during care. In the following quote, a participant reflects on how asking for pronouns made her feel safer.

*Maybe asking people’s pronouns, I feel that it is something very important, because you just... you change your perspective immediately, like... cool... although I am a cis woman, I still find it a good thing... you feel like, very safe* (E10).

## 4. Discussion

Based on the experiences and perceptions of the participants in this study, multiple barriers and facilitators that may affect help-seeking were observed. Some of these barriers are explicitly related to LGBT+ issues, such as fear of discrimination or accessing specialised services on LGBT+ issues. Regarding their experiences with mental health care, the perceived effectiveness of services was closely related to accessing safe/affirming care. Differences were found among LGBT+ students, with trans students reporting more barriers and negative experiences with professionals than their cisgender peers. The perception of university mental health services as safe spaces for LGBT+ students was closely related to the positive perception of the university regarding LGBT+ issues. These results show that LGBT+ youth, in addition to the traditional barriers to accessing mental health services during this stage of life, face specific barriers due to their SOGI [46]; furthermore, they highlight the need for safe/affirming mental health services.

### 4.1. Mental Health Help-Seeking: Barriers and Facilitators

In relation to mental health help-seeking, the results revealed barriers at three levels (individual, social, and service). Some of these barriers are explicitly related to LGBT+ issues, including fear of discrimination, concerns about confidentiality of care, negative experiences with professionals, low availability and high cost of specialised services, and denial of care based on trans gender identity. Except for concerns about confidentiality, these barriers have been reported in other studies with non-university LGBT+ populations [14,21,42,47], so they may be independent of age. In contrast, confidentiality of care is related to fear that the professional could disclose the participants’ SOGI to their families, which may be specific to this age group in the Latin American context, where most students continue living with their parents during their time at university. In addition, a wide range of barriers related to attitudes and beliefs about mental health care were observed, which was not the case for social and service-related barriers. This coincides with what has been reported in the literature on the importance of attitudinal barriers in university students [48].

Like barriers, the reported facilitators of help-seeking operated at the individual, social, and service level. The recognition of psychological distress or symptoms by students, family members, or health professionals is relevant for mental health help-seeking, as are positive attitudes towards mental health care [49]. It is not clear whether there are any differences between LGBT+ and HC people in terms of their perceived need for mental care or attitudes towards help-seeking and how this impacts on the use of mental health services [23]. All in all, this mechanism may partly explains the greater use of services by LGBT+ populations, taking into account factors other than their higher prevalence of mental health problems.

### 4.2. LGBT+ University Students’ Experiences and Perceptions about Mental Health Service Use

Accessing safe and affirmative services or, on the contrary, negative experiences related to SOGI during mental health care (including discrimination and microaggressions), are relevant elements that affect the perceive effectiveness of care and satisfaction with mental health services. These experiences are related to Andersen’s concept of effective access [50] which refers to whether the use of services improves users’ health or satisfaction with services. Thus, aggressions and microaggressions during mental health care [23,24] or creating a safe and affirming environment [25,26] can be considered barriers or facilitators of effective access, respectively. In this sense, barriers are not only present while accessing services, but also during mental health care. Like barriers to help-seeking, these types of barriers and facilitators that influence effective access operate in addition to the traditional barriers that affect university students.

One important result regarding university mental health services is that students manifest a favourable attitude towards these services, with some viewing them as safe spaces for LGBT+ people, which could be linked to the generalized perception that this particular university is a safe space for LGBT+ people. This university is known for promoting anti-discrimination and gender recognition policies, even before a gender recognition law had been promulgated in Chile. A study in the United States reported that LGBQ students preferred outside services possibly to obtain affirmative care [14]. In this sense, this study suggests that the positive image of the university regarding LGBT+ issues may contribute to the perception that its health services are safe spaces for LGBT+ students. These results could help to decrease inequity in access to safe services for LGBT+ students, in a context where experiences of discrimination or difficulties in accessing specialised LGBT+ services are common. However, information about these services and their perceived low availability should be considered when promoting the use of these services, as these were barriers for accessing university mental health services reported by the students.

Importantly, barriers to help-seeking and negative experiences in mental health care were mostly reported by trans, non-binary, and gender-fluid students, suggesting that differences may exist within the LGBT+ community itself, with trans people having more difficulties accessing mental health services and experiencing more aggressions and microaggressions by mental health professionals. This could be a reflection of entrenched cissexism and transphobia in society, which manifest themselves in multiple contexts, such as mental health services. One of the consequences of discrimination in mental health care is that those who need professional help in the future may not seek it because of these experiences [51], thus exacerbating gaps in care. It is advisable to explore the differences among LGBT+ populations in future studies and to adopt a range of strategies to address these inequalities in health care.

Although some participants expressed the desire to talk about their SOGI in mental health care sessions, this topic was not always covered: some participants preferred not to do so for fear of suffering discrimination, professionals did not ask, participants considered that it was not relevant because it was not their reason for consulting, or they had concerns over confidentiality, the latter being especially relevant in online mental health care. The literature points out that disclosing SOGI in health care settings may be important for some people, with the subsequent reaction of professionals also playing a key role [52]. Negative reactions may lead to a change of professional, lead the client to conceal their SOGI in future care, or decrease the likelihood of professional help-seeking in the future [52,53]. In this regard, creating safe spaces that can facilitate the disclosure of SOGI can have important effects in reducing health gaps for LGBT+ people [52]. Due to the pandemic, many students had to shift from face-to-face to online mental health care, which resulted in another barrier to disclosing their SOGI to professionals or discussing this topic with them, as others in their home may overhear them. Therefore, creating safe and confidential spaces to discuss sensitive topics is also an important challenge for online mental health care with LGBT+ students.

The recommendations that the participants made to improve the work of professionals and services were mainly aimed at creating safe spaces for LGBT+ people and ensuring that professionals are trained or have knowledge about LGBT+ issues. With respect to professionals, most participants considered that a positive attitude towards LGBT+ people to be important for help-seeking. All of these aspects can be addressed through an affirmative approach during mental health care [28]. The affirmative competencies of professionals, i.e., the attitudes that promote respectful and positive care for LGBT+ persons, are closely linked to levels of education and training in LGBT+ issues [54]. Therefore, it is necessary to promote education efforts aimed at in-service and future mental health professionals to provide culturally competent services for this group [55].

### 4.3. Limitations and Implications

One of the limitations of the study is that participants were recruited in a single public university, which may hinder the transferability of the results to other contexts such as private universities or universities with low acceptability of LGBT+ people. Another limitation is that most of the participants had already used mental health services prior to this study. LGBT+ students with mental health problems but who have never sought help could be a hard-to-reach sample affected by different barriers to help-seeking, which opens new research questions for future studies.

Regarding the implications of the study, it is important to consider that LGBT+ students seek mental health services more than their HC peers [5,14,15,16], however, they must face other barriers associated with LGBT+ identity to access these services, in addition to the traditional barriers affecting university students [46]. The identification of factors that either hinder or facilitate help-seeking can be useful for university and health interventions and policies aimed at promoting mental health help-seeking in this population. Also, the barriers and facilitators to effective access show that it is necessary to continue working on the implementation of safe and affirmative services for LGBT+ people, for example, through the education and training of health professionals in LGBT+ issues [28,54].

The results presented in this article contribute to the literature by describing factors that influence help-seeking and the use of mental health services, two topics that have been less explored for this group. The heterogeneity of the sample allows us to learn about a range of similar and different experiences and issues associated with the students’ SOGI. The results of this study may be useful for planning future research that focuses on the help-seeking experiences and effective access to care of LGBT+ university students. Based on these findings, it would be advisable to explore other dimensions of the phenomenon, such as the intersection between LGBT+ identity and other axes of inequality (e.g., rurality or ethnicity), which could provide valuable findings for improving the accessibility and effectiveness of mental health services.

## 5. Conclusions

Multiple studies have shown that LGBT+ university students use more mental health services than their HC peers, but several barriers limiting access to mental health care remain which have not been explored in depth. The results of this study show that LGBT+ university students face multiple barriers and facilitators that may affect help-seeking. Some of these barriers are explicitly related to LGBT+ issues (e.g., fear of discrimination or difficulties accessing specialised services) and operate in addition to the traditional barriers affecting university students. Effective access to mental health services is closely related to the attitude/response of professionals towards students’ SOGI (e.g., experiences of discrimination or accessing safe/affirming services). Differences were found among LGBT+ students, with trans students reporting more barriers to help-seeking and negative experiences with mental health professionals. With respect to university mental health services, their image as safe spaces for LGBT+ students was closely related to the positive perception of the university regarding LGBT+ issues. These results can be useful for university and health interventions and policies aimed at promoting mental health help-seeking in this population. Furthermore, it is necessary to continue working on the promotion of safe and affirmative services for LGBT+ people through the education and training of health professionals in LGBT+ issues.

## Figures and Tables

**Table 1 ijerph-19-16520-t001:** Characteristics of the participants.

Id	Age	Sexual Orientation	Gender Identity	Pronouns	Mental Health Service Use *
E1	18	Lesbian	Non-binary	She, He, They	Currently: psycho-pedagogical university service (study skills).
E2	18	Questioning	Non-binary	She, He, They	Private psychological care during adolescence. Currently: private psychiatric care (MHP).
E3	19	Attraction to women	Non-binary	She, He, They	Multiple private psychological and psychiatric care services since childhood and currently (MHP).
E4	19	Bisexual	Gender fluid	He, They	Multiple public and private mental health services (outpatient, short stay, and emergency) since adolescence. Currently: university psychological care (MHP).
E5	19	Homosexual	Cisgender man	He	Private psychological care during childhood. Currently: private psychiatric care (MHP).
E6	20	Asexual	Cisgender woman	She	Multiple public and private mental health services since childhood. Currently: private psychological care (MHP).
E7	18	Pansexual	Cisgender woman	She	Private psychological care during adolescence. Currently: public psychological and psychiatric care (MHP).
E8	18	Questioning	Non-binary	They	Multiple private mental health services since adolescence. Currently: private psychological and psychiatric care (MHP).
E9	19	Homosexual	Cisgender man	He	Currently: private psychological care (personal issues).
E10	20	Abrosexual	Cisgender woman	She	Private and university psychological care during last year (MHP).
E11	18	Bisexual	Cisgender man	He	Private psychological care during last year and currently (MHP).
E12	20	Bisexual	Cisgender man	He	Private psychological care during last year (MHP).
E13	18	Bisexual	Cisgender man	He	Private and school mental health services since adolescence. Currently: university psychological care (MHP).
E14	19	Bisexual	Cisgender woman	She	Private psychological care during adolescence and currently (MHP).
E15	18	Bisexual	Cisgender woman	She	Multiple public and private mental health services since adolescence. Currently: public psychological care (MHP).
E16	20	Asexual	Non-binary	He, They	Multiple public, private, and university mental health services (outpatient and hospitalization) since adolescence. Currently: public psychological and psychiatric care, plus private LGBT+ specialised psychological care (MHP).
E17	23	Bisexual	Transgender man	He	Currently: private psychological (HRT) and psychiatric care (MHP).
E18	21	Heterosexual	Transgender woman	She	Private psychological care during childhood (socialization and school adaptation).
E19	19	Bisexual	Transgender man	He	Private psychological care during childhood (gender identity) and adolescence (HRT). Experiences with public mental health services experiences (MHP).
E20	19	Asexual	Non-binary	He, They	Two private psychological care services during adolescence. Currently: private LGBT+ specialised psychological care (MHP).
E21	21	Pansexual	Feminine non-binary	She, He, They	Multiple private, school, and university mental health services since childhood. Currently: private LGBT+ friendly psychological care (MHP).
E22	21	Heterosexual	Transgender woman	She	Multiple private psychological care since childhood (MHP and HRT). Currently: private LGBT+ friendly psychological care (MHP).
E23	21	Heterosexual	Gender fluid	She, He, They	Multiple public mental health services since adolescence. Currently: private psychological and psychiatric care (MHP).
E24	20	Lesbian	Cisgender woman	She	Private psychological care during adolescence (sexual orientation) and currently (MHP).

Notes: MHP = Mental health problems. HRT = Hormone replacement therapy. * Reasons for consulting are shown in brackets. All mental health care experiences consist in outpatient care unless otherwise specified.

## Data Availability

The data are not publicly available because they contain information that could compromise the privacy of research participants.

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
