# Peer review of "Barriers and Facilitators to Mental Health Help-Seeking and Experiences with Service Use among LGBT+ University Students in Chile"

_ijerph, 2022, doi:10.3390/ijerph192416520_

Round 1

Reviewer 1 Report

To begin I would like to extend my thanks to the authors for undertaking this study on the very important topic of experiences of help-seeking amongst LGBT+ university students in Chile. I raise a few minor points for consideration to strengthen the article further, but wish to emphasise that it is an interesting article, which I look forward to seeing published.

Introduction

1.       You give a comprehensive review of the literature, however I would have liked to know a little more about the Chilean context (in terms of attitudes towards LGBT+ people; attitudes towards mental health; university provision of mental health services) to understand the socio-political and economic environment in which your participants’ experiences were situated. This need not be a long explanations, but a brief 2-3 sentences situating the study would have helped me as a reader to understand why you decided that this particular study needed to be undertaken in this particular context.

Methods

2.       You mention collecting data until the saturation criteria was achieved; could you elaborate a little more on what you mean by this.

3.       There was no reflexivity within the methods section, and I think it would be useful to consider how the researchers reflected on their own influence on the data.

4.       Perhaps somewhat related to point 3, in section 2.4 you mention that you used an “inductive thematic content analysis, i.e. that categories of analysis were derived from the data and not from a previously defined theory”. I would argue that the themes you present in the results section are extremely closely related to the interview questions that you asked and so I wonder if you could say a little more about you analytic technique.

Results

5.       In section 3.1 I found the distinction made between the individual, social and services a little unclear and think that I needed more detail on how you carved out these distinctions in order to better understand these findings.

6.       You used the term ‘psychological orientation’ repeatedly in section 3.4. I would ask that you spell out what is meant by this term or instead change this language for something that is more readily understood.

Minor points on language

7.       You use the term ‘social name’ for trans people’s names twice; I would advise you just use ‘name’ instead.

To conclude, I wish you every luck with this article, and hope that this feedback is helpful in the development of this work.

Author Response

Reviewer 1

Comments and Suggestions for Authors

To begin I would like to extend my thanks to the authors for undertaking this study on the very important topic of experiences of help-seeking amongst LGBT+ university students in Chile. I raise a few minor points for consideration to strengthen the article further, but wish to emphasise that it is an interesting article, which I look forward to seeing published.

Introduction

  1. You give a comprehensive review of the literature, however I would have liked to know a little more about the Chilean context (in terms of attitudes towards LGBT+ people; attitudes towards mental health; university provision of mental health services) to understand the socio-political and economic environment in which your participants’ experiences were situated. This need not be a long explanations, but a brief 2-3 sentences situating the study would have helped me as a reader to understand why you decided that this particular study needed to be undertaken in this particular context.

Reply: We added information specific to the Chilean context at the end of the introduction. We hope this could inform about the socio-political environment in which the study was conducted.

Methods

  1. You mention collecting data until the saturation criteria was achieved; could you elaborate a little more on what you mean by this.

Reply: We described what we meant by data saturation in the manuscript. We consider data saturation when the last interviews repeated what was found in the previous ones.

  1. There was no reflexivity within the methods section, and I think it would be useful to consider how the researchers reflected on their own influence on the data.

Reply: We follow your recommendation. Information about the researchers was added to the procedure section to account for the reflexivity.

  1. Perhaps somewhat related to point 3, in section 2.4 you mention that you used an “inductive thematic content analysis, i.e. that categories of analysis were derived from the data and not from a previously defined theory”. I would argue that the themes you present in the results section are extremely closely related to the interview questions that you asked and so I wonder if you could say a little more about you analytic technique.

Reply: We used and inductive approach to the data because there is not enough knowledge about mental health help-seeking and experiences with mental health service use among college students. We wanted to be open to discovering new elements that might be specific to LGBT+ university students. In contrast, deductive content analysis uses a specific theory to develop an analysis matrix. Then, the data is coded to check correspondence with or exemplification of the identified categories in the analysis matrix. Despite this difference between inductive and deductive content analysis, we know that our knowledge about the research problem and the questions asked in the interview could influence the coding, categorisation, and interpretation of the results. However, it is important to note that we do not used the interview script an analysis matrix.

For the analysis process, we first did an open coding. Two different researchers read each interview separately and took notes on all the themes that emerge from them. These notes were the codes that were then compared and discussed to reach a consensus. Secondly, we grouped the codes into categories based on their similarities. This process was done in order to describe what we found and to broaden our knowledge of the phenomenon. Thirdly, we formulated a description of the categories and named them according to their content (Elo & Kyngäs, 2008, https://doi.org/10.1111/j.1365-2648.2007.04569.x). The results were presented to other members of the research team to discuss them. According to the above, we improve the description of the analysis process in the manuscript.

Results

  1. In section 3.1 I found the distinction made between the individual, social and services a little unclear and think that I needed more detail on how you carved out these distinctions in order to better understand these findings. (MATIZAR).

Reply: We found several barriers and facilitators to help-seeking. To improve the presentation of the results, we classified the barriers according to who or what - primarily – prevented/delayed or promote help-seeking. Based on the reviewer's comment, we improved the description of this classification in the manuscript.

  1. You used the term ‘psychological orientation’ repeatedly in section 3.4. I would ask that you spell out what is meant by this term or instead change this language for something that is more readily understood.

Reply: The term “psychological orientation” was change to “psychological care”. It was a translation error. We are sorry for the mistake and thank you for pointing it out. The final manuscript was edited by a professional English editor to avoid these issues.

Minor points on language

  1. You use the term ‘social name’ for trans people’s names twice; I would advise you just use ‘name’ instead.

Reply: Thank you for pointing this out. We follow your advice in the results section, but in two places we change “social name” by “updated name” (lines 57 and 122) to keep the main idea of the sentences. We hope this correction is more appropriate.

To conclude, I wish you every luck with this article, and hope that this feedback is helpful in the development of this work.

Reply: We appreciate your comments. We sincerely believe that the comments of both reviewers have strengthened the manuscript.

Author Response

Reviewer 2

Comments and Suggestions for Authors

This study reports the qualitative interview results of (N = 24) LGBTQ university students from the same institution in Chile. The manuscript aims to identify barriers and facilitators for LGBTQ university students to access university counseling services. Although there is some literature on barriers and facilitators to mental health care among the LGBTQ general population, there is a current gap in the literature focused specifically on university counseling settings. The methods are well-described, and the sample and data collected are appropriate to answer the research question. I have both major and minor points to consider before recommending this to publication.

Major points

  1. Overall, the results are detailed and have an appropriate balance of describing themes and exemplary quotes, however, I’m unsure the usefulness of organizing the themes by ‘SOGI related’ or ‘not SOGI related,’ as this seems to be based on whether SOGI was at all mentioned in an readily apparent way, but does not consider how larger structures of heterosexism, misogyny, cissexism/transphobia, transmisogyny, etc. may play a role in participants experiences or themes that may ‘apply to all university students’ For example, lines 305-315 all barriers reported here were by trans-identified and cisgender female participants, which may signal to the role of a patriarchal system influencing participants’ experiences. This is acknowledged in the discussion, but I think perhaps warrants additional attention. In another example, lines 227-244, participants noted reaching out to their parents for help to access mental health care. Although this is categorized as ‘non-SOGI related’ I wondered about whether participants were out to their parents, and how this could also affect the parent-child relationship, and comfort asking for help. If the authors thought this organization to be particularly useful for recommendations to providers or institutions, perhaps they can expand on that. Otherwise, it seems a superficial way to organize themes that may hinder the overall presentation of results.

Reply: We thank the reviewer for their comments. The classification into "SOGI related" and "non-SOGI related" was used to present the results more clearly. After a long discussion among the authors, we agree with the reviewer on this issue. Consequently, we changed the description of the results. In some cases we highlighted the results as "explicitly related" to SOGI. By doing so, we wanted to differentiate that these particular results (e.g., discrimination by professionals based on SOGI) are specific among LGBT+ university students. In relation to reaching out to their parents for help, this was expressed by different students regardless their disclosure status. However, we agree with the reviewer in this point. Multiple cultural aspects of society (such as heterosexism or cissexism) are implicit.

  1. The current organization of the discussion makes it unclear what contributions the current findings make specifically to understanding the university counseling context, versus findings/themes already described in the current literature on LGBTQ populations broadly. The content appears in the discussion, but gets a bit lost. Perhaps a re-organization with subheadings may be useful to understand what is unique about the university context, what was found in the current study that is new/has not been described in prior literature, and what themes were consistent with other literature outside of the university context.

Reply: Based on the reviewer’s comment, we modified the discussion and use subheadings to highlight the main findings. This implied that we reorganize some of the contents of the article to respond to the research objectives and to highlight the most important findings in the abstract, discussion, and conclusion. We hope that his rearrangement could help to highlight the main findings of the study.

  1. The authors may consider providing information on the interviewers identities, roles and what was known by the participants. Please refer to COREQ guidelines domain 1 on research team and reflexivity: https://academic.oup.com/intqhc/article/19/6/349/1791966

Reply: We follow your recommendation. Information on the investigators was added to the procedure section.

  1. It is notable that participants described experiences of sexual orientation and gender identity change efforts (SOGICE) or so called “conversion therapy” This is acknowledged once on line 410, but perhaps there is an opportunity to name this as such (i.e., SOGICE) when it is mentioned on lines 297-303 and 375 as well. I am aware of a professional organization working on a systematic review of the harmful/iatrogenic effects of SOGICE, so naming this explicitly may help in future policy work to outlaw these harmful practices.

Reply: Thank for your comment. We change both paragraphs mentioned.

Minor points

  1. Paragraph starting on line 52 seems like it could be combined with the paragraph starting on line 57, as both discuss barriers and facilitators in LGBT university students and the LGBT general population. Perhaps consider reorganizing the two paragraphs separated by sexual minority and gender minority.

Reply: We change the beginning of the paragraph to better differentiate the main idea of both paragraphs. The paragraph beginning on line 52 gives an account of experiences with mental health help-seeking. The paragraph starting on line 57 describe the experiences during mental health care (i.e., once they have already accessed mental health services).

  1. What is meant by “psychological orientation” on line 297? Perhaps this phrasing was lost in translation.

Reply: It was change to “psychological care”. It was a translation error. We are sorry for that mistake, and we thank you for pointing it out. The final manuscript was edited by a professional English editor to avoid these issues.

We appreciate your comments. We sincerely believe that the comments of both reviewers have strengthened the manuscript.
